# In Vitro Effects of Leaf Extracts from *Brassica rapa* on the Growth of Two Entomopathogenic Fungi

**DOI:** 10.3390/jof7090779

**Published:** 2021-09-19

**Authors:** Daniel G. Cerritos-Garcia, Pasco B. Avery, Xavier Martini, Valentina Candian, Liliana M. Cano, Ronald D. Cave

**Affiliations:** 1Indian River Research and Education Center, Department of Entomology and Nematology, Institute of Food and Agricultural Sciences (IFAS), University of Florida, 2199 South Rock Road, Fort Pierce, FL 34945, USA; dgc4@illinois.edu (D.G.C.-G.); rdcave@ufl.edu (R.D.C.); 2Department of Agricultural Sciences and Production, Zamorano University, San Antonio de Oriente, Fco. Morazán, Honduras; 3North Florida Research and Education Center, Department of Entomology and Nematology, Institute of Food and Agricultural Sciences (IFAS), University of Florida, Quincy, FL 32352, USA; xmartini@ufl.edu; 4Indian River Research and Education Center, Department of Plant Pathology, Institute of Food and Agricultural Sciences (IFAS), University of Florida, 2199 South Rock Road, Fort Pierce, FL 34945, USA; valentina.candian@unito.it (V.C.); lmcano@ufl.edu (L.M.C.); 5Department of Agricultural, Forest and Food Sciences (DISAFA), University of Torino, 2 Largo P. Braccini, 10095 Grugliasco, Italy

**Keywords:** liquid nitrogen, radial growth, *Cordyceps fumosorosea*, *Beauveria bassiana*, leaf extracts, stimulatory, gas chromatography–mass spectrometry analysis, sesquiterpenes

## Abstract

This study aimed to determine the inhibitive or stimulatory effects of leaf extracts from two *Brassica rapa* subspecies on the hyphal growth of two well-known entomopathogenic fungi, *Cordyceps fumosorosea* and *Beauveria bassiana.* Extract concentrations of 50, 25, and 10% *w*/*v* based on leaf fresh weight were prepared from turnip (*B. rapa* subspecies *rapa*) and bok choy (*B. rapa* subspecies *chinensis*) leaves. Each concentration was individually incorporated into potato dextrose agar plates for in vitro bioassays. The center of each plate was inoculated with 20 µL of a fungal suspension that was allowed 24 h to soak into the agar before sealing the plates and incubating them at 25 °C under a 14-h photophase. The fungal colony perimeter was marked 5 days after inoculation on two perpendicular lines drawn on the bottom of each plate. Radial colony growth was measured from 4 marks per plate 5, 10, and 15 days later. Radial growth rates for both fungi were 1.3–2.0 and 0.9–1.4 times faster with bok choy and turnip extracts, respectively, at the 25% and 50% concentrations compared to the no-extract control treatment. Therefore, bok choy and turnip leaf extracts can stimulate entomopathogenic fungus growth within 15 days. Biochemical compounds in the extracts include sesquiterpenes, α-copaene, β-selinene, γ-gurjunene, calamenene, cubenene, and α-calacorene.

## 1. Introduction

The Brassicaceae, commonly known as crucifers, have been studied extensively for secondary metabolites, known as glucosinolates, found exclusively in their tissues [1]. When the plant is mechanically damaged, such as being fed on by an arthropod or infected by a plant pathogen, cellular breakdown exposes the glucosinolates stored in the leaves to degradative enzymes known as myrosinases [2]. The products of the glucosinolate enzymatic hydrolysis, known as isothiocyanates, are reported to have nematicidal [3,4], fungicidal [5,6], and bactericidal [6,7,8] properties. In addition, the sequestration of these toxic isothiocyanates in the hemocoel of arthropod pests can negatively influence the efficacy of EPF used for biological control [9,10,11].

Crucifer crops are an important source of food for many cultures around the world. The genus *Brassica*, with approximately 159 species, is considered the most important of this family because it includes many economically important crops [11], such as kale, cabbage, bok choy, cauliflower, broccoli, turnip, rapeseed, and canola. Production of these cruciferous crops is often difficult, and yield is significantly reduced by arthropod pests such as the yellowmargined leaf beetle, *Microtheca ochroloma* Stål, especially in organic production [12]. The yellowmargined leaf beetle is a specialist herbivore capable of causing significant economic damage to organic crucifer crops throughout the southeastern United States [13,14]. Important management tactics for this pest in organic production include predators, trap cropping using turnip, botanical insecticides, and entomopathogenic fungi (EPF), which can be employed alone or in combination [12,13,14,15,16,17,18].

The advantages of using EPF-based biopesticides are their compatibility with natural enemies and pollinators, their residue-tolerant exempt status, and their ability to be incorporated as a more ecologically sustainable tactic in integrated pest management programs compared to using broad-spectrum synthetic chemical pesticides that are not selective and can be toxic to non-target organisms [19]. EPF-based biopesticides are now commercially available on the market worldwide and can be safely handled by the applicator with little risk of toxic effects in the case of drift [20,21,22,23,24,25,26]. The unique attribute of EPF-based biopesticides is that the pest herbivore can become infected in two ways: (1) ingestion of spores *per os* with food material; and (2) integument penetration after acquiring the propagules as the pest roams on the contaminated leaf surface. The propagules will adhere to the pest insect, germinate, form hyphae, then differentiate into appressoria with a penetration peg, and via pressure and specific enzymes will penetrate through the integument until it reaches the hemocoel and forms hyphal bodies called blastospores [27,28,29].

If, during feeding, the herbivore sequesters host plant biochemicals that are inhibitory or antagonistic to the growth of the EPF, the insect could potentially be protected from an infection in the hemocoel. Poprawski and Jones [30] observed that the mycosis of whiteflies (*Bemisia* sp.) reared on cotton (*Gossypium hirsutum* L.) were significantly less susceptible to infection by *Paecilomyces fumosorosea* (Wize) Brown and Smith (now *Cordyceps fumosorosea* (Wize) Kepler, B. Shrestha and Spatafora [31] and *Beauveria bassiana* (Balsamo) Vuillemin) compared to whiteflies reared on melon (*Cucumis melo* L). They hypothesized that gossypol produced by the cotton plant may confer protection to the whiteflies from infection by the EPF after feeding on and possibly sequestering the allelochemical from the plant. After assessing in vitro studies testing gossypol for germination inhibition of the EPF, they concluded that sequestered gossypol (and other plant allelochemicals) in the whiteflies would help to partially explain the insect’s defense against the insect pathogens. In a similar study, Tian et al. [32] assessed the susceptibility of *Bemisia tabaci* (Gennadius) to *C. fumosorosea* reared on various plants (cucumber, eggplant, tomato, and bean) and concluded that host plants affect the pathogenicity and virulence of an herbivore pathogen. Poprawski et al. [33] evaluated the susceptibility of third instars of the greenhouse whitefly, *Trialeurodes vaporariorum* (Westwood), reared on cucumber or tomato plants to infection by *C. fumosorosea* and *B. bassiana*. They concluded that nymphs reared on tomato plants were significantly less susceptible to the EPF compared to nymphs reared on cucumber and attributed this antimicrobiosis partially to the glycoalkaloid tomatine sequestered from the tomato leaves.

Multitrophic plant–pest–entomopathogen interactions are complex and can be difficult to assess. Designing studies to evaluate the efficacy of an EPF as a potential biocontrol agent of a specialist arthropod feeding on a host plant that contains several allelochemical compounds can be especially difficult. Therefore, interpretation of the resultant data needs to be evaluated carefully with this complexity in mind and is dependent on whether the study was conducted in vitro, in vivo, or both. For example, Vega et al. [34] demonstrated in vitro that the pure isothiocyanate sinigrin in cabbage (*Brassica oleracea* L.) leaves may inhibit the germination and subsequent fungal growth of *Metarhizium anisopliae* (Metchnikoff) Sorokin and *C. fumosorosea*. Inyang et al. [35] observed that isothiocyanates inhibited the in vitro growth of *M. anisopliae*, but in vivo the pathogenicity of this EPF against the mustard beetle, *Phaedon cochleariae* (Fabricius), feeding on Chinese cabbage was affected by various crucifer-derived stimulatory and inhibitory compounds. Klingen et al. [36] showed that 100 ppm of 2-phenylethyl isothiocyanate completely inhibited the growth of *M. anisopliae* and *Tolypocladium cylindrosporum* Gams in vitro; however, when using an in vivo fungus–plant–soil microcosm system, no fungal inhibition was found. Ujjan and Shahzad [37] observed in vivo that some fungal strains of *M. anisopliae*, *Lecanicillium lecanii* (Zimmerman) Zare and Gams, *B. bassiana*, and *Purpureocillium lilacinum* (Thom) Luangsa-ard, Hywel-Jones, Houbraken, and Samson, were efficacious against the mustard aphid, *Lipaphis erysimi* (Kaltenbach), after topical applications in the laboratory and under field conditions. Lastly, Dos Anjos et al. [38] observed in vivo that yellowmargined leaf beetles feeding on crucifer leafy greens were susceptible to infection by *B. bassiana* under field conditions.

The ingestion and sequestration of allelochemical substances by arthropods that consume the brassicaceous plant tissue could play a role in these specialist herbivores rarely becoming infected by EPF [11,27,28]. Gámez Herrera et al. [17] and Montemayor et al. [18], who investigated the efficacy of commercially formulated *C. fumosorosea* applied against yellowmargined leaf beetles on bok choy (*Brassica rapa* subspecies *chinensis*) plants, observed in vivo, that an application rate 2× the label rate resulted in only 35% and 4% mortality of first instars and adults, respectively. Therefore, it appears that the efficacy of *C. fumosorosea* may be hindered due to the sequestration of glucosinolates by the beetle from feeding on the bok choy leaves. In addition, considering that the preferred host plant of the yellowmargined leaf beetle is turnip, which has been employed as a trap crop in an integrated management strategy for this pest [11,13,14,16], we decided to investigate the compatibility of *C. fumosorosea* on turnip as well. Moreover, because Dos Anjos et al. [38] observed that *B. bassiana* could infect this leaf beetle, the compatibility of commercially available fungal products containing different strains of *B. bassiana* with both bok choy and turnip leaf extracts was determined.

Therefore, to better understand this plant–insect–entomopathogen interaction, the objective of our study was to first determine in vitro if the leaf extracts from *B. rapa* subspecies *rapa* (turnip) and *B. rapa* subspecies *chinensis* (bok choy), which may be sequestered from feeding on the leaf tissue, have a stimulatory or an inhibitive effect on the hyphal growth of *C*. *fumosorosea* and *B. bassiana*. Additionally, the biochemicals found in the leaf extracts were identified and compared with a gas chromatography–mass spectrometry (GC-MS) analysis to determine if there are any differences in their composition.

## 2. Materials and Methods

### 2.1. Fungal Species and Fungus-Based Products Used in This Study

The experiment used the following products as treatments: (1) PFR 97™ 20% WDG (PFR-97; Certis Biologicals, Columbia, MD, USA), a powdered, wettable, dispersible, granular blastospore formulation containing the Apopka strain of *C. fumosorosea* as the active ingredient (a.i.); (2) a powdered blastospore formulation containing ARSEF 3581 (ARSEF), a Texas strain of *C. fumosorosea*, as the a.i. (USDA, ARS, NCAUR, Peoria, IL, USA); (3) BotaniGard^®^ ES (BotaniGard; Laverlam International, Butte, MT, USA) an oil conidial formulation containing GHA strain of *B. bassiana* as the a.i.; and (4) BioCeres WP (BioCeres; BioSafe Systems, East Hartford, CT, USA), a wettable powder conidial formulation containing ANT-03 strain of *B. bassiana* as the a.i. All products were refrigerated at 4 °C until use, except BotaniGard ES that was stored on the laboratory bench at room temperature (23–24 °C). A viability test was performed for all fungal strains prior to inoculations, and a threshold of >90% of viable spores was obtained.

### 2.2. Plant Material

Turnip and bok choy seeds were sown in 72-cell trays containing Fafard^®^ Super-Fine Germinating Mix (Sun Gro Horticulture, Agawam, MA, USA) for at least 20 days. After this, seedlings were transplanted to 3.8 L black plastic pots containing SunGro^®^ Professional Growing Mix (Sun Gro Horticulture, Agawam, MA, USA). The plants were watered daily and fertilized with Osmocote^®^ 15:9:12 controlled-release encapsulated formulation (Scotts Miracle-Gro Co., Marysville, OH, USA) for 3 months (Figure 1a,b).

### 2.3. Leaf Extract Preparation

Twenty-five grams of leaf material were collected 50 days after planting. Extracts from each plant subspecies were prepared by using a modified protocol from Naidu [39]. For this protocol, individual leaves (wet mass) were placed in a mortar, and liquid nitrogen was added as needed. Leaves were crushed and ground with a pestle into a fine powder. From the resultant leaf powder, 12.5 g were placed into a sterile 50 mL propylene Corning^®^ CentriStar™ centrifuge tube (Corning Inc., Corning, NY, USA) and used immediately or stored at −80 °C until ready for transport for chemical identification. For immediate usage in fungal growth assays, 25 mL of 80% ethanol was added to the powder, and the mixture was allowed one hour for extraction of the chemicals. After this time, the extract was vortexed for 15 sec, filtered through two layers of sterile gauze, and centrifuged at 4000 rpm for 15 min at 4 °C to remove any cell fragments. The resultant extract supernatant was used to produce 50 mL of a 50% *w*/*v* concentration based on the fresh weight of the leaves. From this 50% concentration solution, 25 and 5 mL (30 mL total) of the supernatant were removed and diluted with 25 mL and 45 mL of ethanol, respectively, in separate tubes to produce a 25% and a 10% *w*/*v* concentration, respectively (Figure 1c). All supernatant concentrations were then immediately incorporated into the potato dextrose agar (PDA) (Difco™, Benton, Dickinson and Co., Sparks, MD, USA) plates to be used for in vitro bioassays.

### 2.4. Effect of Leaf Extracts on Fungal Growth

Liquid medium (19.5 g of PDA powder/500 mL of sterile distilled water) was mixed in a 1 L bottle by using a magnetic stirring bar and then autoclaved for 20 min at 120 °C. After the liquid medium cooled to ~16 °C, 50 mL of supernatant from one of the ethanol- leaf extract concentrations were added and the medium was mixed again with a magnetic stirrer for 1 min. Liquid medium (22 mL) containing a specific leaf extract concentration was poured into each of 128 sterile plastic Petri dishes (100 mm × 15 mm) and allowed to solidify. For the control, 50 mL of 80% ethanol were added to the liquid media prior to pouring. Perpendicular lines traversing the diameter of the agar plate (Figure 1e–g) were drawn on the underside of the bottom plate with colored permanent markers prior to pouring the agar.

Prior to inoculation, *C. fumosorosea* (PFR 97™ 20% WDG and ARSEF 3581) and *B. bassiana* (BotaniGard^®^ ES and BioCeres^®^ WP) suspensions were made from the formulated products mixed with sterile, distilled water and adjusted to a concentration of 10⁶ spores/mL. The PDA plates were inoculated by placing a 20 µL drop of the fungal suspension on the center of the plate (Figure 1d) at the intersection of the two perpendicular lines, and 24 h were given for the suspension to soak into the agar before sealing the Petri dish with Parafilm™ M (Bemis Co., Neenah, WI, USA) and placing it in a growth chamber set at 25 °C with a 14 h photophase. There were five replicates per fungal suspension per leaf extract concentration for both plant subspecies. For each fungal suspension per plant subspecies treatment, there was a control treatment, as described above, with five replicates.

The perimeter of the fungal colony was marked on each perpendicular line on the underside of each plate 5 days after inoculation, and the radius was measured. Fungal colony growth was assessed 5, 10, and 15 days later by measuring the colony’s radius from the crossing of the perpendicular lines to the perimeter of the colony in two perpendicular directions. Radial growth (RG) per time period was determined by the following formula
RG = (R_x_ − R_0_);(1)
where R_x_ = colony radius in mm at 5, 10, or 15 days after the initial fungal colony radial measurement; R_0_ = initial fungal colony radius in mm 5 days after inoculation. The purpose of this measurement technique was to avoid bias in radial growth measurements caused by variability in colony establishment during the first 5 days. The experiment was conducted twice on separate occasions. When working with entomopathogens, especially fungi, it is necessary to conduct the experiment at least twice or more times to confirm results. One sample from the same product batch could give different results considering it is a living organism.

### 2.5. Gas Chromatography–Mass Spectrophotometry Analysis of Leaf Extracts

Using a scoop previously dipped in liquid nitrogen, leaf powder (5 g) stored at −80 °C was carefully placed into Corning^®^ Centistar™ 50 mL sterile centrifuge tubes, capped tightly, and left floating in a container with liquid nitrogen until they could be packed for shipping. Tubes were capped tightly, caps sealed with Parafilm, and immediately packed into a styrofoam container with dry ice and shipped overnight from the University of Florida (UF), Indian River Research and Education Center in Ft. Pierce, FL overnight to the UF North Florida Research and Education Center in Quincy, FL to be analyzed. Upon arrival, the leaf powder samples were stored at −20 °C until ready for chemical identification with gas chromatography–mass spectrometry (GC-MS) analysis. Before analysis, leaf samples were defrosted at 21 °C for 3 h, then, 10 mL of dichloromethane was added to each tube. Tubes were centrifuged at 13,000 rpm for 1 min, and 1 mL of supernatant was diluted in dichloromethane at a ratio of 1:10,000. One µL of the diluted solution was then injected into an ISQ QD Single quadrupole GC-MS spectrophotometer (Thermo Fisher Scientific, Inc. Waltham, MA, USA). The gas chromatograph was equipped with a TG-5MS capillary column (30 mm × 0.250 mm inner Ø; 0.25 µm film thickness, Thermo Fisher Scientific, Inc. Waltham, MA, USA). Data collection, storage, and subsequent analysis were performed on Thermo Fisher Chromatographic data system Chromeleon 7™ Thermo Fisher Scientific, Inc. Waltham, MA, USA). Helium at a linear flow velocity of 1 mL/min was used as the carrier gas. The temperature of the column oven was maintained at 40 °C for 1 min and then increased at a rate of 10 °C/min to a final temperature of 250 °C and maintained at 250 °C for 1 min. The injector temperature was set at 270 °C. Constituents of the plant volatile were identified by comparison of mass spectra with spectra in the National Institute of Standards and Technology database, the Flavors and Fragrances of Natural and Synthetic Compounds (FFNSC 3, Wiley, Hoboken, NJ, USA) and the spectra obtained from authentic reference compounds, when available. Additionally, GC retention times of plant volatiles were compared with those of authentic compounds on the TG-5MS column, when available. For each compound, the relative area was calculated by dividing the area of the compound peak with the sum of all the areas of the selected peaks. There were three replicates for each plant species.

### 2.6. Data Analysis

Mean in vitro radial growth measurements at 15 days were compared statistically with Student’s *t*-test (α = 0.05). A best fit linear regression model analysis (α = 0.05) was used to compare the effect of the extracts on the radial growth slopes of each fungal entomopathogen over time. Differences among treatment means of the in vitro linear growth for each fungal entomopathogen per each concentration of the same plant extract were compared statistically using an ANOVA and *post-hoc* separation of the mean values by a Tukey’s HSD test (α = 0.05). Statistical analyses were conducted using SAS 9.4 for WINDOWS 2012 (Cary, NC, USA) and gpplot2 library from R software version 4 [40].

## 3. Results

### 3.1. Effect of Leaf Extracts on Fungal Growth

Significant differences in fungal colony radial growth at 15 days were detected among treatments for all four fungal strains on PDA with bok and turnip leaf extracts. For PFR-97, radial growths with 25% and 50% concentration of bok choy leaf extract were similar but significantly greater (*F*_3, 67_ = 10.50; *p* < 0.0001) than on the control treatment (Figure 1e and Figure 2). Growth on the 25% extract concentration was significantly greater than growth on the 0% and 10% concentrations. There was no significant difference between the 10% concentration and the control treatment. Radial growth on 50% concentration of turnip leaf extract was significantly greater (*F*_3, 67_ = 9.27; *p* < 0.0001) compared to the other three concentrations (Figure 2).

Radial growth of ARSEF 3581 at 15 days was significantly greatest (*F*_3, 67_ = 144.96; *p* < 0.0001) on 50% bok choy leaf extract concentration compared to the other concentrations and significantly least on the control treatment (Figure 2). Radial growth was significantly greatest (*F*_3, 67_ = 78.00; *p* < 0.0001) on 25% turnip leaf extract concentration and significantly least on 0% and 10% extract concentrations (Figure 2).

For BioCeres WP, radial growth at 15 days on PDA with bok choy leaf extracts was significantly greater (*F*_3, 67_ = 21.15; *p* < 0.0001) with all concentrations compared to the control (Figure 1f and Figure 2). Radial growth with 50% extract concentration was significantly greater than with 25% but similar to growth at the 10% concentration. Radial growth on the turnip leaf extract at 25% and 50% concentrations was similar and significantly greater (*F*_3, 67_ = 9.75; *p* < 0.0001) than in the control treatment, but only radial growth on the 50% concentration was significantly different from the growth on the 10% concentration (Figure 2).

For BotaniGard ES, radial growth on all bok choy extract concentrations was significantly greater (*F*_3, 67_ = 16.43; *p* < 0.0001) compared to the control treatment (Figure 2). Radial growth on the 50% extract concentration was significantly greater than with the 10% concentration but similar to growth on the 25% concentration treatment. On turnip leaf extracts, radial growth was significantly greater (*F*_3, 67_ = 6.82; *p* = 0.0004) on the 25% concentration compared to the 0% and 50% concentrations, but it was similar to radial growth with 10% extract concentration (Figure 1g and Figure 2). No difference in radial growth was observed between the 50% concentration and control treatments.

The slopes of the growth models for PFR-97, ARSEF, BioCeres, and BotaniGard on 50% bok choy leaf extract concentration were significantly steeper ((*t*_118_ = −2.95; *p* = 0.0039; *n* = 60), (*t*_118_ = −8.15; *p* < 0.0001; *n* = 60), (*t*_118_ = −3.28; *p* = 0.0014; *n* = 60), and (*t*_118_ = −3.34; *p* = 0.0011; *n* = 60), respectively) than those of their control treatment models (Figure 3). ARSEF with 25% turnip leaf extract concentration and BioCeres with 50% turnip leaf extract concentration grew significantly faster ((*t*_118_ = −4.66; *p* < 0.0001; *n* = 60) and (*t*_118_ = −2.26; *p* = 0.0258; *n* = 60), respectively) compared to their control treatments (Figure 4). The growth model slopes for PFR-97 with 50% concentration and BotaniGard with 25% turnip leaf extract concentration were not significantly different ((*t*_118_ = −1.44; *p* = 0.1536; *n* = 60) and (*t*_118_ = −1.85; *p* = 0.0673; *n* = 60), respectively) from those of the control treatments (Figure 4).

Comparisons between fungal strains are based on the extract concentration at which fungal growth was fastest. On PDA with 50% bok choy extract, mean radial growths of ARSEF, PFR-97, and BioCeres at 15 d were similar among themselves but significantly greater (*F*_3, 57_ = 86.26; *p* < 0.0001) than BotaniGard (Table 1). On PDA with 50% turnip extract, radial growths of PFR-97 and BioCeres at 15 d were similar yet significantly greater (*F*_3, 57_ = 131.06; *p* < 0.0001) than those of ARSEF and BotaniGard with 25% turnip extract (Table 1).

The slopes of the growth models for PFR-97 with 50% concentration and ARSEF with 25% concentration were similar when grown on PDA with extracts from bok choy (*t*_118_ = 1.80; *p* = 0.0750; *n* = 60) and turnip (*t*_118_ = −1.22; *p* = 0.2665; *n* = 60) (Figure 5). The slope of the radial growth model was significantly steeper for BioCeres with 50% concentration than for BotaniGard with 25% concentration when grown on PDA incorporated with extracts from bok choy (*t*_118_ = −5.63; *p* < 0.0001; *n* = 60) and turnip (*t*_118_ = −7.72; *p* < 0.0001; *n* = 60) (Figure 5).

### 3.2. Gas Chromatography–Mass Spectrophotometry Analysis of Leaf Extracts

We identified 17 volatile compounds from the turnip and bok choy leaf extracts (Figure 6). Major biochemical compounds were α-copaene, β-selinene, γ-gurjunene, calamenene, cubenene, and α-calacorene. For both source plant subspecies, the major compound was calamenene, a sesquiterpene accounting for 22.9% in bok choy extract and 38.0% in turnip extract of the total profile. Overall, bok choy extracts were very rich in sesquiterpenes as they represented >95% of the compounds identified, while turnip extracts had only 80% of sesquiterpenes. Compounds found in turnip but not (or at very low concentration) in bok choy were cymene, undecane, and dodecane. While statistical analyses were not conducted on these data due to the small number of replicates, the standard errors of the means of cymene, dodecane, undecane 4,7 dimethyl, and α-calacorene did not overlap, thus indicating potential differences for these compounds between turnip and bok choy.

## 4. Discussion

Our study demonstrates that bok choy extracts in vitro are stimulatory to the hyphal growth of *C. fumosorosea* and *B. bassiana* strains within 15 d compared to the addition of no extract (Figure 2 and Figure 3). Radial growths at day 15 for all EPF subjected to 50% concentration of bok choy and turnip leaf extracts were greater than radial growth on PDA without leaf extract; however, radial growths of *C. fumosorosea* Apopka strain and *B. bassiana* GHA strain exposed to the turnip leaf extract were greater at the 25% concentration level (Figure 3). Although interesting, these findings are contrary to what we expected to observe based on a previous in vivo study where the yellowmargined leaf beetle adult was rarely infected by *C. fumosorosea* Apopka strain while feeding on bok choy leaves sprayed with the fungus [17,18]. However, in corroboration with the findings in our study, Lin et al. [41,42] observed that the herbivore-induced plant volatiles (HIPVs) emitted from the crucifer *Arabidopsis thaliana* (L.) Heynh, while its leaves were fed on by the aphid *L. erysimi*, were stimulatory to spore germination and appressorial formation of *L. lecanii*, which are requisites for pathogenicity of the fungal infection.

Our study is novel and the first to assess the hyphal growth of *C. fumosorosea* and *B. bassiana* strains exposed to leaf extracts of bok choy and turnip in vitro at different concentration levels. However, similar in vitro research that preceded our study varied greatly in protocol, EPFs tested, or both. For example, Vega et al. [34] used the same ARSEF strain of *C. fumosorosea*, but their protocol differed from our study: (1) Germination of the blastospores of *C. fumosorosea* was the focus, not radial growth of the fungal colony; (2) Blastospores were grown on Noble agar, not PDA; (3) Rather than whole leaf extracts, sinigrin, a biochemical found in crucifers, was incorporated into the Noble agar. Vega et al. [34] observed that sinigrin did inhibit blastospore germination. Inyang et al. [35] studied in vitro the effect of volatile isothiocyanates from Chinese cabbage (*Brassica campestris* var. *pekinensis*) on the radial growth of *M. anisopliae* on Sabouraud dextrose agar plates. Instead of incorporating an isothiocyanate into the agar, a drop of the biochemical was placed on the inner surface of the Petri dish lid to simulate an HIPV-effect on the EPF. In their study, the isothiocyanates negatively affected spore germination, hyphal development, and the ability of the EPF to infect adult *P. cochleariae*. Klingen et al. [36] investigated the potential interactions between rutabaga (*Brassica napus* L.), the isothiocyanates they produce upon wounding (simulated by grating), and *M. anisopliae* and *T. cylindrosporum*. They concluded that isothiocyanates can inhibit the growth of EPF in vitro, but after using a more realistic in vivo tritrophic microcosm, no fungal inhibition was observed. In our study, the use of liquid nitrogen to freeze the leaf tissue during pulverization may have allowed the HIPVs inside the leaf powder to be preserved and may have meant that the isothiocyanates usually activated upon wounding or feeding were not released. Moreover, by using leaf extracts that contain a whole composite of interacting biochemicals rather than an isolated isothiocyanate incorporated into the agar, the in vitro condition simulated in our study may be more reflective of the realistic in vivo conditions affecting the EPF on the leaf surface.

The effect that exudates and HIPVs can have on the growth and pathogenicity of an EPF will be either inhibitive, neutral, or stimulatory. In our in vitro study, with bok choy leaf extract incorporated in the PDA, the stimulatory effect on the EPF increased as the concentration of the extract increased. However, the effect of the turnip leaf extract on the hyphal growth of each EPF was very different and not always stimulatory at the highest concentration level. Hyphal growth of *C. fumosorosea* Apopka strain in PFR-97 was stimulated to grow faster when exposed to the turnip extract only at the 50% concentration level, but the lower concentrations had a neutral effect. In contrast, hyphal growth of the *C. fumosorosea* ARSEF strain was stimulated more at the 25% concentration level compared to the 10% concentration, which had a neutral effect, whereas the effect on hyphal growth was inhibitive or slowed at the 50% concentration level. A stimulatory effect on hyphal growth for the *B. bassiana* ANT-03 strain in BioCeres was also observed when this strain was exposed to the turnip extract. The stimulatory effect from turnip extract on *B. bassiana* GHA strain resulted in a similar trend to that of the *C. fumosorosea* ARSEF 3581 strain, except that at the highest concentration level the effect was neutral.

Our observation of a stimulatory effect on the EPF hyphae to grow faster as the bok choy extract concentration increased may reflect a similar HIPV-effect as reported by Lin et al. [41,42], who observed that as the number of aphids feeding on *A. thaliana* leaf increased, the concentration of HIPVs increased, and, subsequently, the germination and appressorial formation of *L. lecanii* increased as well. Therefore, from an ecological perspective, the release of the HIPVs from the leaf would stimulate the EPF to grow faster, ultimately infecting and killing the feeding herbivore faster as well. This idea of fungal entomopathogens serving the role of “bodyguards” on the leaf phylloplane was proposed by Elliot et al. [43], Cory and Hoover [44], and Cory and Ericsson [45]. Based on our findings, it is possible that the crucifer leaf “attacked” by herbivores will release HIPVs to stimulate EPF growth to “help” kill the herbivore feeding on it. Although speculative and interesting, this hypothesis requires further study to confirm this specific scenario. However, the yellowmargined leaf beetle was observed to be unaffected by the presence of *C. fumosorosea* propagules after feeding on bok choy leaf [17,18]. Researchers investigating similar tritrophic interactions have concluded that plant exudates and HIPVs released on the leaf phylloplane can affect: (1) the impact of insect fungal pathogens by enhancing or decreasing persistence on the leaf surface [35,41,46], (2) the encounter rate between the insect and the pathogen [47], (3) insect susceptibility to disease [33,48,49,50,51,52].

The major biochemical compounds in bok choy and turnip leaf extracts are α-copaene, β-selinene, γ-gurjunene, calamenene, cubenene, and α-calacorene. Sesquiterpenes are known to be major compounds found in *Brassica* species [53]. Nondestructively collected turnip leaf volatiles are known to include mainly sesquiterpenes but also terpenes and green leaf volatiles [54]. It is possible that the pulverization of the leaves increased the release of sesquiterpenes from the leaf tissues. Interestingly, some sesquiterpenes are known to induce hyphal branching in arbuscular mycorrhizal fungi [55] and the growth of the ascomycete *Ophiostoma floccosum* Math.-Käärik [56]. Therefore, it is possible that sesquiterpenes found in turnip and bok choy extracts might be implicated in the observed increased hyphal growth of the EPF. However, this hypothesis requires further study to be confirmed. In addition, sesquiterpenes are also known to be produced by some fungi for a possible defensive role [57].

## 5. Conclusions

It is important to determine the potential effect of a host plant’s secondary allelochemicals on the growth of an EPF contained in a fungal biopesticide product in vitro, prior to it being applied under greenhouse or field conditions, in order to maximize the efficacy of any fungal biopesticide. In our case, the in vivo study was conducted first where we observed that the yellowmargined leaf beetle feeding on the bok choy leaves was difficult to induce mortality with *C. fumosorosea* applied at 2× the label rate [17,18]. We hypothesized that the growth and subsequent pathogenicity of the fungus may be negatively influenced by the presence of glucosinolates in the hemocoel sequestered *per os* from the leaves during beetle herbivory. However, contrary to our previous in vivo observations, results from this in vitro study revealed that, in general, the leaf extracts of bok choy and turnip when incorporated in the agar were stimulatory, not inhibitory, to the hyphal growth of the EPF. Based on the design of our in vitro study, the positive hyphal stimulation was due to the increase in the added biochemical nutrients contained in the extract incorporated in the agar, which was absent in the control plates. The GC-MS analysis of the leaf extracts identified many different biochemicals that may have played a role in stimulating the hyphal growth of the EPF tested, including *C. fumosorosea*, previously applied as a fungal biopesticide in the prior in vivo study [17,18]. This same scenario concerning the influence of the host plant on the performance of the EPF after application on different leaf surfaces has been repeated by other researchers as well where the in vivo study preceded the in vitro study [33,41,42]. For example, Lin et al. [41] first determined in vivo that the HIPVs induced by multiple aphids feeding on the brassica plant leaves were stimulatory to the conidial performance of *L. lecanii* and enhanced the pathogenicity against the aphids. A later study conducted by Lin et al. [42] determined that of all the HIPVs released during herbivory by the aphids, only the biochemicals methyl salicylate and menthol positively affected the growth and pathogenicity of *L. lecanii*. Therefore, further analysis using purified compounds, particularly the sesquiterpenes, should be conducted to determine which of the biochemicals contained in the complex extract of *B. rapa* subspecies has a positive effect on the growth of each EPF tested in this study. Confirmation of the effect of sequestered leaf extracts of both *B. rapa* subspecies on the efficacy of the fungal biopesticides should be assessed under field in vivo conditions.

## Figures and Tables

**Figure 1 jof-07-00779-f001:**
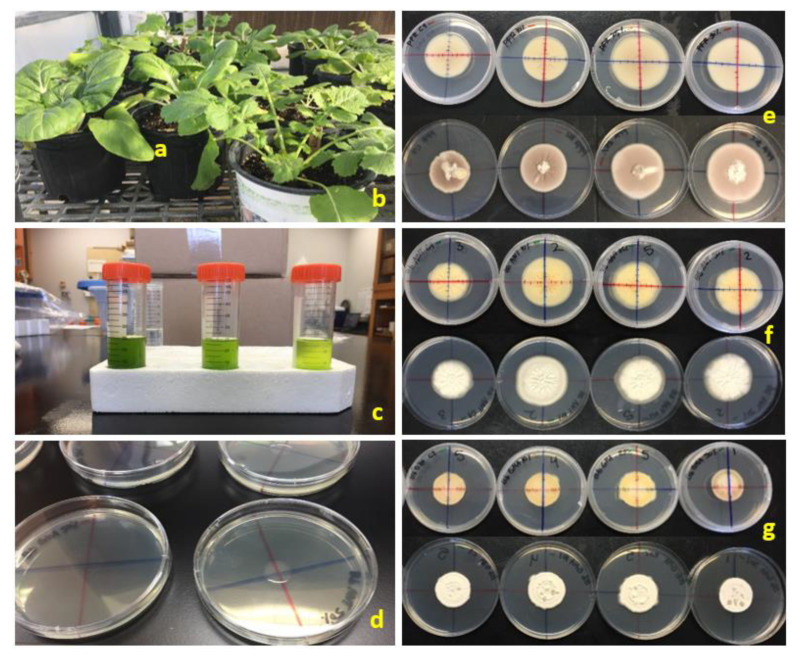
Study preparation and sample results after exposing in vitro two entomopathogenic fungi to leaf extracts from two *Brassica rapa* subspecies. Materials: (**a**) bok choy; (**b**) turnip; (**c**) supernatant extract concentrations in 50 mL tubes (left to right: 50, 25, and 10%); (**d**) 20 µL inoculation droplet placed in the center of the PDA plate. Radial fungal growth on plates after exposure to 0, 10, 25, and 50% (left to right) plant extract concentrations for 15 days: (**e**) *Cordyceps fumosorosea* (PFR-97—strain Apopka) with bok choy extract; (**f**) *Beauveria bassiana* (BioCeres—strain ANT-03) with bok choy extract; (**g**) *Beauveria bassiana* (BotaniGard—strain GHA) with turnip extract. Top plate sequences in photographs (**e**–**g**) are of the dish underside with colored perpendicular lines drawn for measuring radial fungal growth over time. Bottom plate sequences in photographs (**e**–**g**) are of the dish topside.

**Figure 2 jof-07-00779-f002:**
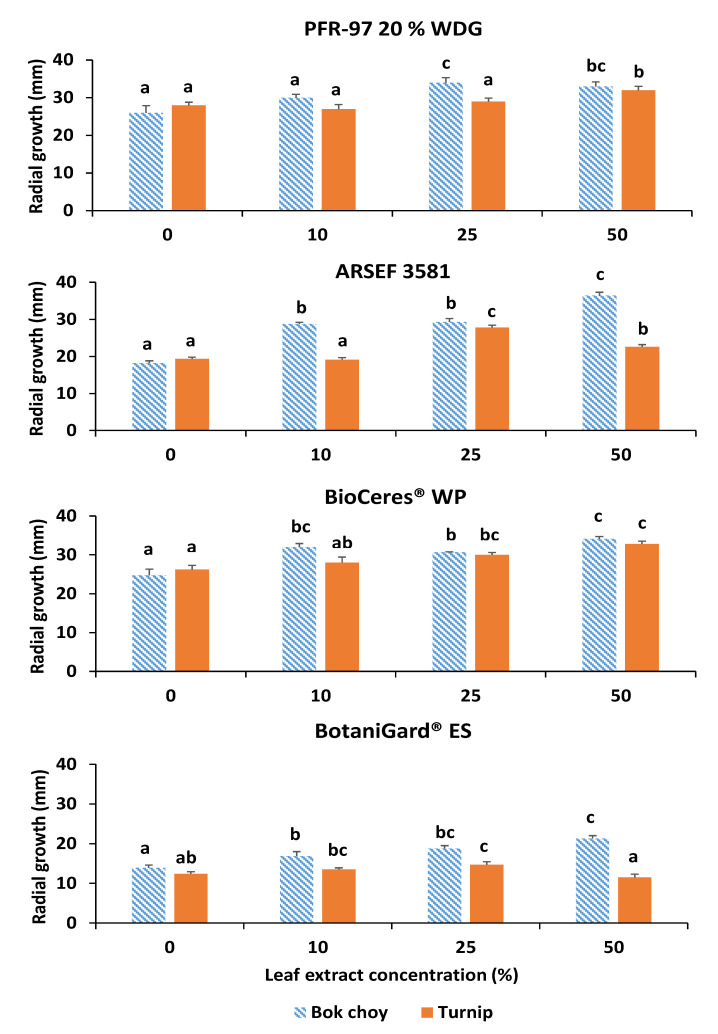
Mean (±SE) 15 d fungal radial growth (mm) from four entomopathogenic fungal products grown on PDA with four leaf extract concentrations from bok choy or turnip. For each product, bars for each leaf extract plant source (same color) with different letters are significantly different (Tukey’s HSD test, *p* < 0.05).

**Figure 3 jof-07-00779-f003:**
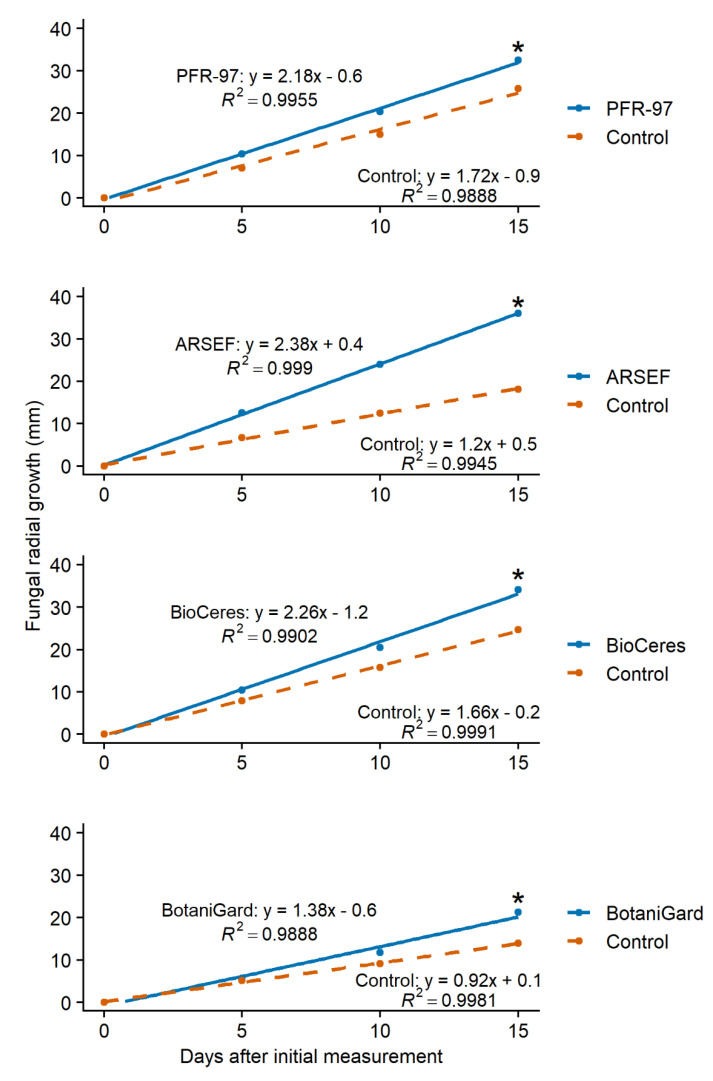
Linear models of fungal radial growth for four entomopathogenic fungal products grown on PDA with 50% concentration of bok choy leaf extract compared to its control treatment model over a 15-day observation period. Asterisk (*) above a line endpoint represents significance between treatment and control linear models (Student’s *t*-test, *p* < 0.05).

**Figure 4 jof-07-00779-f004:**
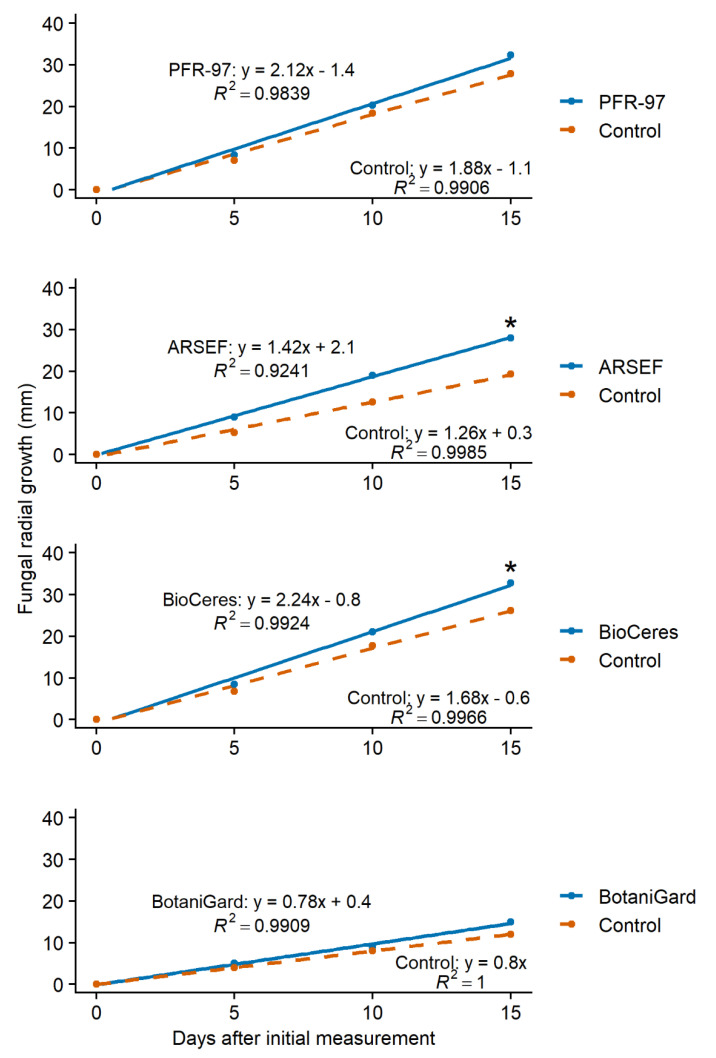
Linear models of fungal radial growth for four entomopathogenic fungal products grown on PDA with 50% (PFR-97 and BioCeres) or 25% (ARSEF and BotaniGard) concentration of turnip leaf extract compared to its control treatment over a 15-day observation period. Models for PFR-97 and BioCeres are based on 50% leaf extract concentration; models for ARSEF and BotaniGard are based on 25% because radial growth was fastest at this concentration. Asterisk (*) above a line endpoint represents significance between treatment and control linear models (Student’s *t*-test, *p* < 0.05).

**Figure 5 jof-07-00779-f005:**
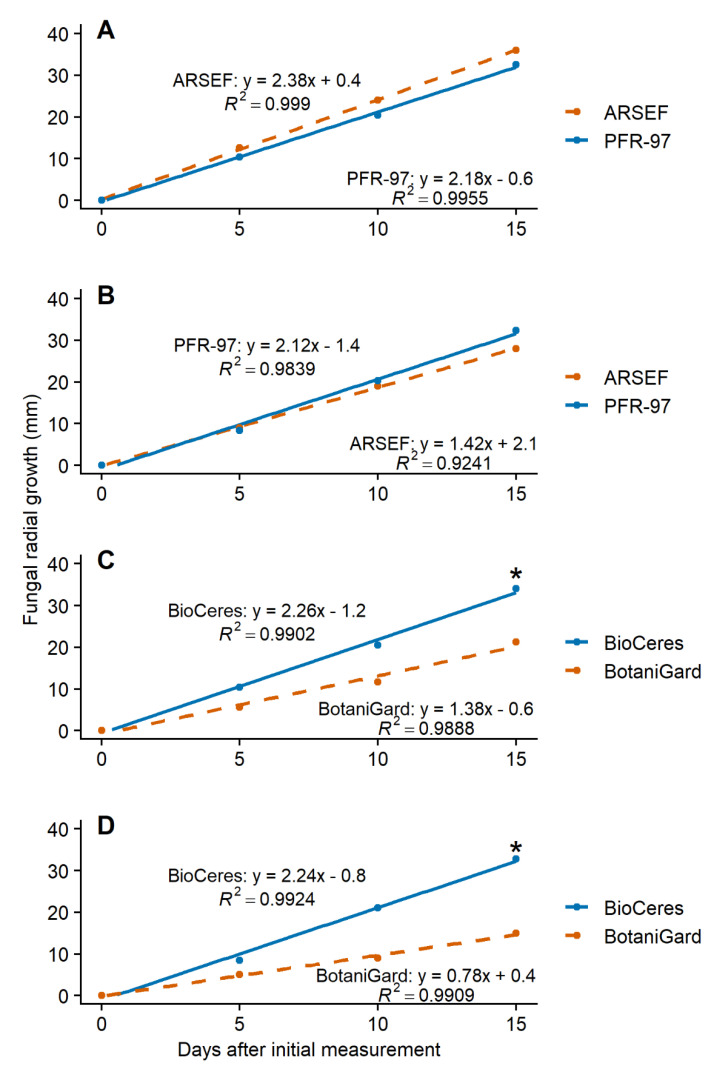
Linear models of fungal radial growth over 15 days that compare fungal products of the same species grown on PDA with bok choy (**A**,**C**) or turnip (**B**,**D**) leaf extract. Models for PFR-97 and BioCeres are based on 50% leaf extract concentration; models for ARSEF and BotaniGard are based on 25% because radial growth was fastest at this concentration. Asterisk (*) above a line endpoint represents significance between treatment models (Student’s *t*-test, *p* < 0.05).

**Figure 6 jof-07-00779-f006:**
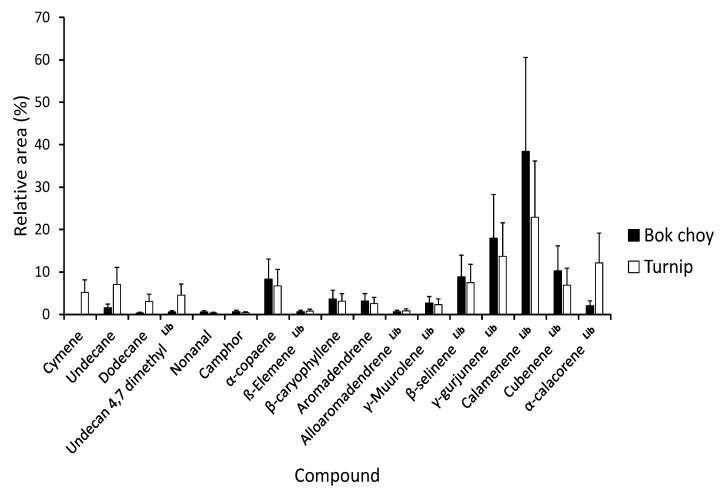
Mean (± SE) relative area for the 17 compounds in volatiles from powdered bok choy and turnip leaves. Compounds are presented in the order of elution. ^Lib^ (superscript) indicates compounds identified based on libraries only, as standards were not available.

**Table 1 jof-07-00779-t001:** Comparison of the mean (± SE) fungal radial growth (mm) at 15 days of four entomopathogenic fungal products on PDA with either bok choy or turnip leaf extracts.

		Mean Radial Growth ^a^	
Fungal product	Species (strain)	Bok choy ^b^	Turnip ^c^
PFR-97	*Cordyceps fumosorosea* (Apopka)	33 ± 1.2 ^b^	32 ± 1.0 ^c^
ARSEF 3581	*Cordyceps fumosorosea* (3581-TX)	36 ± 0.9 ^b^	28 ± 0.6 ^b^
BioCeres WP	*Beauveria bassiana* (ANT-03)	34 ± 0.6 ^b^	33 ± 0.7 ^c^
BotaniGard ES	*Beauveria bassiana* (GHA)	21 ± 0.7 ^a^	15 ± 0.7 ^a^

^a^ Values in the same column not followed by the same letter are significantly different (Tukey’s test, *p* < 0.05). ^b^ Radial growth means for all fungal strains are from 50% leaf extract concentration. ^c^ Radial growth means for PFR-97 and BioCeres WP are from 50% leaf concentration; ARSEF 3581 and BotaniGard ES are from 25% concentration (growth was greatest at these concentrations).

## Data Availability

The data presented in this study are available on request from the corresponding author.

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
