# Peer review of "In Vitro Effects of Leaf Extracts from Brassica rapa on the Growth of Two Entomopathogenic Fungi"

_jof, 2021, doi:10.3390/jof7090779_

Round 1

Reviewer 1 Report

All the previous comments have been addressed by the authors. The decision is accepted for publication.

Comments for original manuscript  jof-1135492

This work is settled in the exciting research field of multitrophic plant-pest-entomopathogen interactions, and the finding is a quite important contribution to this field. I am recommending accepting the manuscript with minor revision. In general, the manuscript is well organized and careful written. However there are some minor points that ought to be ascertained by the authors.

Specific comments:

Line 146. Please include the concentration of streptomycin and chloramphenicol.

Line 159. Why are the fungi exposed to 14-h photophase?

Figures 1 and 2. Place letters in each graph as in figure 3

Figures. Place the four graphs and the food of figures in the same page

Figure 3. Why are models of PFR and ARSEF based on 50% and BioCeres and BotaniGard based on 25% leaf concentration?

Table 1. Idem

Line 351. “...was not easily infected by the PFR-97...” There should be other factors related to the penetration process influencing the successful of the infection. Deep on this idea briefly with new references.

Author Response

Comments and Suggestions for Authors

This work is settled in the exciting research field of multitrophic plant-pest-entomopathogen interactions, and the finding is a quite important contribution to this field. I am recommending accepting the manuscript with minor revision. In general, the manuscript is well organized and careful written. However, there are some minor points that ought to be ascertained by the authors.

Specific comments:

Line 146. Please include the concentration of streptomycin and chloramphenicol.

Author response: Because the 80% ethanol was already in the extract, there was no need to add the two bactericides streptomycin (0.1%) and chloramphenicol (0.1%) to the PDA liquid medium prior to pouring the plates. The mention of bactericides has been deleted from the text.

Line 159. Why are the fungi exposed to 14-h photophase?

Author response: Our reference strain of Cordyceps fumosorosea PFR 97, also known as Apopka, was originally isolated in Apopka, Florida. Therefore, in our bioassay we used the 14-h photophase characteristic of Florida in summer.

Figures 1 and 2. Place letters in each graph as in figure 3

Author response: We disagree. The name of the fungal product is already indicated in each graph labelled next to the appropriate regression line and below the horizontal axis. Only Figure 3 has letters to distinguish between the different extracts.

Figures. Place the four graphs and the food of figures in the same page

Author response: I moved the text to allow the graphs to be all on the same page; however, I had to move the text which is now several pages ahead of Figures 2-3.

Figure 3. Why are models of PFR and ARSEF based on 50% and BioCeres and BotaniGard based on 25% leaf concentration?

Author response: That is incorrect. The models with the turnip extract for ARSEF and BotaniGard were based on 25% concentration level because the growth rate was faster than at 50%. The growth of all fungi exposed to the bok choy extract was based on 50% concentration. I clarified this in all the text and figure captions related to ARSEF and BotaniGard.

Table 1. Idem

Author response: same as above

Line 351. “...was not easily infected by the PFR-97...” There should be other factors related to the penetration process influencing the successful of the infection. Deep on this idea briefly with new references.

Author response: Extra information was added in this section with citations also included in the references list

Reviewer 2 Report

Dear Authors after reviewing your ms with title In Vitro Effects of Leaf Extracts from Brassica rapa on the Growth of Entomopathogenic Fungi, have not understand fully the prepose of your Ms.

My major concern is the idea of this experiment. You say at the lines 96-97 to better understand the plant insect EPF interaction. How, using only the plant extract? The influence of the extract how does you except to work?

My decision is reject because the idea of the experiment and the use of the commercial strains is not accepted.

Author Response

Comments and Suggestions for Authors

Line 65 give the scientific name of the cabbage

Author response: done- incorporated: cabbage (Brassica oleracea L.)

Describe the control treatments instead of indicating that "for each fungal suspension per plant subspecies treatment, there was a control treatment"

Author response: Revised and described in paragraph.

Line 171- Could you explain the separate occasions and what informed your decision to conduct the experiment twice?

Author response: When working with entomopathogens, especially fungi, it is necessary to conduct the experiment at least twice or more times to confirm your result. One sample from the same batch could give differing results considering it is a living organism.

184 give the actual temperature at the time of the experiment

Author response: 21 °C; incorporated in text

DATA ANALYSIS
Rewrite the first sentence under data analysis to make sense -Line 206 -207.

Author response: Rewritten as follows: “Mean in vitro radial growth measurements at 15 days were compared statistically with Student’s t-test (α = 0.05).”

RESULTS 
The vertical axis on figure 1 label is not visible/readable

Author response: I tried to fix it without the captions or graphs going to the following page, but the text that had preceded the graphs had to be moved to the beginning of the section to fill-in the space taken up by Figure 1. The editor will need to remedy this situation.

DISCUSSION
The discussion needs to be precise and well-articulated. In as much as other studies are an important reference in supporting the novelty of this study, the authors have talked more about other works and findings as opposed to discussing their findings. 

I recommend that they beef up the discussion part to capture more of their finding.

Author response: We have added two paragraphs discussing our findings and their relevance to this study and added many new references to support my statements. 

Reviewer 3 Report

The results shown in the manuscript are interesting. It includes a phytochemical part that gives more value to the discussion carried out by the authors.

Author Response

Comments and Suggestions for Authors

The results shown in the manuscript are interesting. It includes a phytochemical part that gives more value to the discussion carried out by the authors.

Author response: We are glad you found it interesting… thank you!

Reviewer 4 Report

Manuscript title: In Vitro Effects of Leaf Extracts from Brassica rapa on the

The manuscript “In Vitro Effects of Leaf Extracts from Brassica rapa on the

Growth of Entomopathogenic” is a decent piece of work. However, there are some major pointes to be rectified in order to make the manuscript for further consideration

  1. Abstract seems to be well written.
  2. Introduction section should be included about the advantages of microbial pathogen please do refer the previously published papers doi: 1016/j.jip.2018.10.008; doi:10.1007/978-81-322-2056-5_3
  3. P3-L95-101. Please specify the relation between leaf extract and endomopathogenic fungi with relevant supporting research.
  4. P3-L-106 Please include more detail on the cultures of the tested fungi, especially the formulation-application over the media in which the EPF grown.
  5. Most the experiment groups lack control. Please add control group with proper statistical analysis
  6. Second paragraph of introduction shall be moved above – starting section of introduction to maintain a coherent flow.
  7. Discussion section is very long and it should be condensed A clear correlation between obtained results and previous citations should be illustrated. Discussion section should be re written.
  8. Accordingly, the bibliography should be drastically modified

Though the authors have validated their research in a very capable way, substantial evidences of at least couple of photographs could have been added. The manuscript shall be considered only after major revision.

Author Response

Comments and Suggestions for Authors

Manuscript title: In Vitro Effects of Leaf Extracts from Brassica rapa on the Growth of Entomopathogenic

The manuscript “In Vitro Effects of Leaf Extracts from Brassica rapa on the Growth of Entomopathogenic” is a decent piece of work. However, there are some major pointes to be rectified in order to make the manuscript for further consideration

  1. Abstract seems to be well written.

      Author response: Thank you!

  1. Introduction section should be included about the advantages of microbial pathogen please do refer the previously published papers doi: 1016/j.jip.2018.10.008 (Kumar et al. 2019); doi:10.1007/978-81-322-2056-5_3 (Senthil-Nathan 2015)

Author response: Advantages of microbial pathogens are now included, with the above publications incorporated as well as other authors.

  1. P3-L95-101. Please specify the relation between leaf extract and entomopathogenic fungi with relevant supporting research.

Author response: The relationship between the leaf extract and entomopathogenic fungi with relevant supporting research is specified.

  1. P3-L-106 Please include more detail on the cultures of the tested fungi, especially the formulation-application over the media in which the EPF grown.

Author response: Detail on the cultures and formulation-application over the media has been included.

  1. Most the experiment groups lack control. Please add control group with proper statistical analysis

Author response: We disagree with the reviewer’s statement that the experiment lacks control treatments. Moreover, statistical analyses did include the control. Figures 1-4 include the control (0 extract added to the PDA). Only Figure 5 does not include the control because it is not relevant to what is being conveyed in the graph.

  1. Second paragraph of introduction shall be moved above – starting section of introduction to maintain a coherent flow.

Author response: Second paragraph has been moved above to precede first paragraph.

  1. Discussion section is very long and it should be condensed. A clear correlation between obtained results and previous citations should be illustrated. Discussion section should be re written

Author response: Discussion section is rewritten and condensed as requested.

  1. Accordingly, the bibliography should be drastically modified

Author response: The number of references is reduced.

  1. Though the authors have validated their research in a very capable way, substantial evidence of at least couple of photographs could have been added.

Author response: I created a photo plate that becomes Figure 1. This includes photos of the methodology and a sample of the different concentrations for commercialized biopesticides.

The manuscript shall be considered only after major revision.

Round 2

Reviewer 2 Report

Dear Authors,

this a very good paper now. 

Author Response

Dear Authors,

This a very good paper now. 

Author response: Thank you!

Reviewer 4 Report

The authors have done a good job

Author Response

Dear Authors,

The authors have done a good job 

Author response: Thank you!

This manuscript is a resubmission of an earlier submission. The following is a list of the peer review reports and author responses from that submission.

Round 1

Reviewer 1 Report

This work is settled in the exciting research field of multitrophic plant-pest-entomopathogen interactions, and the finding is a quite important contribution to this field. I am recommending accepting the manuscript with minor revision. In general, the manuscript is well organized and careful written. However there are some minor points that ought to be ascertained by the authors.

Specific comments:

Line 146. Please include the concentration of streptomycin and chloramphenicol.

Line 159. Why are the fungi exposed to 14-h photophase?

Figures 1 and 2. Place letters in each graph as in figure 3

Figures. Place the four graphs and the food of figures in the same page

Figure 3. Why are models of PFR and ARSEF based on 50% and BioCeres and BotaniGard based on 25% leaf concentration?

Table 1. Idem

Line 351. “...was not easily infected by the PFR-97...” There should be other factors related to the penetration process influencing the successful of the infection. Deep on this idea briefly with new references.

Reviewer 2 Report

Line 65 give the scientific name of the cabbage
Describe the control treatments instead of indicating that "for each fungal suspension person plant subspecies treatment, there was a control treatment"

Line 171- Could you explain the separate occasions and what informed your decision to conduct the experiment twice?

184 give the actual temperature at the time of the experiment

DATA ANALYSIS
Rewrite the first sentence under data analysis to make sense -Line 206 -207.

RESULTS 
The vertical axis on figure 1 label is not visible/readable 

DISCUSSION
The discussion needs to be precise and well-articulated. In as much as other studies are an important reference in supporting the novelty of this study, the authors have talked more about other works and findings as opposed to discussing their findings. 

I recommend that they beef up the discussion part to capture more of their finding.